# Elevated Coronary Artery Calcium scores are associated with tooth loss

H. C. M. Donders[1,2☯]*, L. M. IJzerman[3☯], M. Soffner[4], A. W. J. van 't Hof[5,6‡], B. G. Loos[3‡], J. de Lange[1‡]

1 Department of Oral and Maxillofacial Surgery, Amsterdam UMC, Academic Centre of Dentistry Amsterdam (ACTA), University of Amsterdam, Amsterdam, The Netherlands, 2 Department of Oral and Maxillofacial Surgery, Isala Hospital Zwolle, Zwolle, The Netherlands, 3 Department of Periodontology, Academic Centre of Dentistry Amsterdam (ACTA), University of Amsterdam and Vrije Universiteit Amsterdam, Amsterdam, The Netherlands, 4 Practice for Endodontology, TSC Hoorn, Hoorn, The Netherlands, 5 Department of Cardiology, Maastricht University Medical Center and Cardiovascular Research Institute Maastricht (CARIM), Maastricht, The Netherlands, 6 Department of Cardiology, Zuyderland MC, Heerlen, The Netherlands

☯ These authors contributed equally to this work.
‡ These authors also contributed equally to this work.
* h.c.m.donders@isala.nl

## Abstract

### Aim

This study explores the association between Coronary Artery Calcium (CAC) scores and dental pathology such as missing teeth, the (peri-apical) health status and restoration grade of the teeth, and the grade of alveolar bone loss seen on a dental panoramic radiograph (Orthopantomograph–OPG).

### Materials and methods

In this retrospective cross-sectional study, data was collected from three hospitals spread in the Netherlands. Patients were included when a CAC score and an OPG were available, both recorded within a maximum period of 365 days from 2009–2017. The CAC score was measured on a CT scan, using the Agatston method. To assess dental pathology, the number of missing teeth, the number of dental implants, alveolar bone loss, caries, endodontic treatments, peri-apical radiolucencies, bone loss at implants, impacted teeth and dental cysts, were determined on the OPG. All observers were calibrated. The electronic health records provided information about: gender, age, smoking, Diabetes Mellitus, hypercholesterolemia, hypertension and Body Mass Index (BMI).

### Results

212 patients were included. We found a statistically significant association between the number of missing teeth and the CAC score. When modeling age, sex, and other well-known risk factors for cardiovascular disease, the significant correlation was no longer present after multivariate correction. Furthermore, the results showed a trend for more teeth with peri-apical lesions and a higher percentage of mean alveolar bone loss in the group with the highest CAC scores.

**Data Availability Statement:** All relevant data are within the manuscript and its Supporting Information files.

**Funding:** The author(s) received no specific funding for this work.

**Competing interests:** The authors have declared that no competing interests exist.

## Conclusion

This study showed that being edentulous or missing teeth is correlated to higher CAC scores however failed to be an independent predictor of atherosclerotic cardiovascular diseases. The number of (missing) teeth is an easily accessible marker and could be used as a marker for atherosclerotic cardiovascular disease (ACVD) risk by almost any healthcare worker. The current study needs to be considered as an explorative pilot study and could contribute to the design of further (prospective) studies on the relationship between dental pathology and coronary artery calcification by adding clinical information and extra cardiovascular biomarkers.

## Introduction

Atherosclerotic cardiovascular diseases (ACVD) are one of the leading causes of death and morbidity in the Western world [1]. The underlying pathology, atherosclerosis, is a progressive disease characterized by the accumulation of lipids and fibrous elements in the arteries. Over the past two decades, inflammation has emerged as an integrative factor for coronary atherosclerosis. Inflammation can be evident in all stages of this disease, from initiation through progression and, ultimately, the thrombotic complications of coronary atherosclerosis [2].

Remarkable epidemiological and pathological associations between oral health and cardiovascular diseases have been reported. The first study that found evidence for the association between dental pathology and coronary heart disease was in 1989 by Mattila et al. [3]. Since then a multitude of cross-sectional and longitudinal studies have implicated periodontitis as a risk for ACVD in addition to the well-known risk factors including smoking, hypercholesterolemia and diabetes [4].

Tooth loss is the ultimate event representing dental pathologies. Various articles have found that missing teeth are predictors of incident CVD, for example as was reported by Liljestrand et al. [5] More recently an interesting meta-analysis on tooth loss and risk of ACVD and stroke was published [6]. Dental caries is a lifelong disease and traditionally considered as an important cause of tooth loss. Dental caries has a multifactorial etiology; consumption of dietary carbohydrates, composition of the oral flora and poor oral hygiene are the most important etiological factors. Frencken et. al determined in their systematic review a global age-standardized prevalence of untreated dentine carious lesions in the permanent dentition of 35% There were no significant differences between sexes and disease prevalence reached its peak at age 25, with a second peak later in life at around 70 years of age [7].

Another dental pathologic condition that can lead to tooth loss is apical periodontitis. This is a chronic inflammation around the apex of a tooth, in most cases caused by bacterial invasion of the pulp and root canal, most often as a result of untreated dental caries. This condition is frequently asymptomatic and may progress with the resorption of apical periodontal ligament and surrounding alveolar bone; some cases flare up, however most peri-apical lesions are discovered on routine dental x-ray's during dental checkups. These peri-apical lesions contain bacteria which can be translocated throughout the body and lodge in various organs and also in atherosclerotic lesions [8, 9]. Nevertheless there are only a few studies that have suggested an association between chronic apical periodontitis and cardiovascular disease [10].

Periodontitis on the other hand, is a chronic multi-causal inflammatory disease of the supportive tissues of the teeth with progressive loss of attachment and alveolar bone, finally

leading to tooth loss [11]. It is the most common oral disease, affecting 30–50% of the adults and approximately 10% of the population in its most severe form [12]. Quite some research has been performed to identify pathophysiological mechanisms to explain the association between periodontitis and coronary heart disease [13]. Recently an update on the association and plausible mechanisms how periodontitis can be a risk factor for ACVD has been published [14].

Since inflammation has emerged as an integrative factor for cardiovascular disease, many studies used biochemical inflammatory biomarkers such as cytokines (IL-6, TNF-α), cell adhesion molecules (P-selectin) and acute-phase reactants (CRP, fibrinogen) as surrogate parameters for cardiovascular risks [15]. However, in addition to inflammatory biomarkers, there are a number of clinical non-invasive surrogate markers of cardiovascular disease. These are related to the endothelial function and arterial stiffness, including measurement of the carotid arteries, echocardiography, ankle-brachial index, flow-mediated dilation (FMD) in the brachial artery and pulse waved velocity analysis [16]. These surrogate cardiovascular biomarkers have been widely used to explore the association between dental pathology and cardiovascular diseases [17–19].

Nowadays, Coronary Artery Calcium (CAC) scoring has emerged as a widely available, consistent and reproducible means of assessing risk for major cardiovascular outcomes, especially useful in asymptomatic people for planning primary prevention interventions [20]. Coronary artery calcium provides superior discrimination and risk reclassification of cardiovascular disease in intermediate-risk individuals, compared with ankle-brachial index, high-sensitivity CRP and family history [21]. CAC scoring has sporadically been used to investigate the association between periodontitis and cardiovascular diseases [22, 23]. The current retrospective cross-sectional pilot study explored the association of Coronary Artery Calcium (CAC) scores with radiographic parameters of dental pathology, including missing teeth, periodontal disease, dental caries and peri-apical disease.

## Materials and methods

For this retrospective cross-sectional study, data were collected from three hospitals on different locations in the Netherlands (Academic Medical Centre, Amsterdam; Isala Hospital, Zwolle; Ziekenhuis Gelderse Vallei, Ede). Patients were included when the hospital data provided a CAC score and a dental panoramic radiograph (Orthopantomograph–OPG), both obtained within a maximum period of 365 days between them, from 2009–2017. All data were anonymized before accessed. This study was approved by the Medical Ethical Committee (15.06107) of the Isala Hospital, Zwolle and accepted by the other participating hospitals. The Medical Ethical Committee waived the requirement for informed consent.

### Patient characteristics

The electronic health records provided information about: sex, age, smoking, diabetes mellitus, hypercholesterolemia, hypertension and body mass index (BMI). When diabetes mellitus, hypercholesterolemia or hypertension were not mentioned in a patient file, but the corresponding medication was available (e.g. metformin and/or insulin, statins and antihypertensive drugs), the patient was scored positively for that disorder. BMI was calculated with the noted height and weight on the day of the CT-scan for the CAC score.

### Coronary artery calcification

Most of the included patients received a CAC CT-scan because of presentation with symptoms suspected for myocardial ischemia. The CAC scan of the heart was rapidly acquired,

prospectively electrocardiogram-triggered and without contrast. The CAC score was quantified using the Agatston method where the area of calcified atherosclerosis (defined as an area of at least 1 mm$^2$ with a CT density >130 Hounsfield units [HU]) is multiplied by a density weighting factor and summed for the entire coronary artery tree using a 2.5 to 3.0 mm slice thickness CT dataset [24].

## Dental pathology

To asses dental pathology, the following markers were evaluated on the OPG: number of missing teeth, number of dental implants, alveolar bone loss, caries, endodontic treatments, peri-apical radiolucencies, bone loss around dental implants (as a sign for peri-implantitis), impacted teeth and dental cysts. A total of 5 observers were involved in assessing the OPGs and calibration was conducted as follows. Observers A and B (2 dentists, trained by a periodontist) scored the number of present teeth, dental implants and the alveolar bone loss. Observers A and B were calibrated by comparing individual scorings of 10 random OPGs. The results were mutually evaluated, to roughly calibrate the two examiners. Subsequently, 10 new OPGs were scored individually and the agreement was statistically determined. This intraclass correlation coefficient was 0.87. After four weeks the intra-examiner reliability was determined. The same 10 OPGs were scored again and compared with the scores from 4 weeks earlier. The intra-examiner reliability was 0.76 for observer A and 0.73 for observer B. According to Fleiss, scores between 0.4 and 0.75 represent fair to good reliability and scores higher than 0.75 represent excellent reliability [25].

Observers C, D and E (2 oral and maxillofacial surgeons and an endodontist) scored caries, restorations, endodontic treatments, peri-apical radiolucencies, bone loss at implants, impacted teeth and dental cysts. Peri-apical radiolucencies (osteolytic lesions) and dental caries were recorded as present or absent without consideration of size [26]. When in doubt, "present" was assigned. To calibrate the observers, twenty OPGs (randomly selected from the database) were scored and the agreement for each variable was statistically determined by calculating a Cohen's Kappa value. The intra-examiner reliability was 0.98 for caries, 0.90 for restorations, 1.0 for endodontic treatments, 0.89 for peri-apical radiolucencies, 0.74 for bone loss at implants, 1.0 for impacted teeth and 0.73 for dental cysts.

**Number of missing teeth.** The number of present teeth was measured by counting all teeth visible on the OPG, including third molars and radices relictae. Pontics of fixed partial dentures and prosthetic dentures were not counted as teeth. The number of missing teeth was calculated by subtracting the number of present teeth from the expected total of 32 teeth. Dental implants were counted individually.

**Alveolar bone loss.** To score the loss of alveolar bone for each tooth, the cemento-enamel junction (CEJ), the alveolar crest and the apex of the root had to be visible. Using a modified Schei ruler, the loss of alveolar bone was measured in tenths of percentages of the root length. In this study the distance representing the biological width was determined at 2 mm on the Schei ruler, based on the used magnification factor of the printed OPGs, instead of the 1 mm in the conventional Schei ruler, used for intra oral radiographs [27, 28]. Both the mesial and the distal sites were measured. The highest score of each tooth was used for analysis. To determine the alveolar bone loss of a tooth, the transparent Schei ruler was placed on a printed OPG with the marking of the biological width at the CEJ landmark, perpendicular to the longitudinal axis of the tooth and was moved until the last radius covered the apex landmark. The amount of alveolar bone loss was then determined by identifying the position of the alveolar crest relative to the markings of the ruler. For teeth decayed or restored beyond the CEJ, the cervical margin of the decay or restoration was used as the CEJ landmark. For dental implants,

the most apical outline of the crown and the apical end of the implant were used as respectively the CEJ and apex landmarks.

## Statistics

Descriptive statistics (mean ± standard deviations [SD] or numbers [%] of subjects) were used to present patient characteristics and clinical findings. The Shapiro-Wilk test was used for the calculation of the normality of distribution of CAC scores. The patients were grouped in tertiles based on the CAC scores. The mean numbers of the dental pathologies scored on the OPG were calculated per group and possible differences between groups were tested by ANOVA. A backward stepwise linear regression model with variables with $p<0.01$ to stay, was applied to explore any contributing dental factor that appeared to have an uni-variate significance with CAC scores in relation to traditional cardiovascular disease risk factors (age, sex, BMI, diabetes, hypertension, hypercholesterolemia, smoking). For the latter analysis, CAC scores were log transformed to better approach normality of data distribution. Analyses were performed using IBM SPSS Statistics 26 software (SPSS Inc., Chicago, IL, USA). P-values $<0.05$ were considered significant.

## Results

We retrieved 212 patients with an available CAC score and an OPG both recorded within a maximum period of 365 days between them. In 121 (57.1%) patients, the CAC score was assessed before the OPG, in 89 (42.0%) patients the OPG was first available and in 2 (0.9%) patients the CAC score and OPG were taken at the same day. The mean intermediate period between these two radiographic investigations was 170 days (SD 127 days).

The background characteristics of these patients are presented in Table 1. The population of this study consisted of 54% (n = 114) male patients. The mean age was 57.8 years (SD 12.2 years). The mean BMI was 28 kg/m$^2$ (SD 4.9 kg/m$^2$). Fifteen percent (n = 32) of the patients were diabetic, 40% (n = 85) of the patients suffered from hypercholesterolemia, and 60% (n = 128) of the patients were treated for hypertension. The smoking status and smoking history for all patients was divided into three categories: 41% (n = 86) of the patients had never smoked, 41% (n = 86) of the patients were past-smokers and 18% (n = 39) of the patients were current smokers. There was no information available about the period of time the past smoker patients had been smokers (pack-years).

**Table 1. Patient characteristics.**

|  | Total n = 212 | CAC Tertile 1 n = 70 | CAC Tertile 2 n = 70 | CAC Tertile 3 n = 72 |
|---|---|---|---|---|
| Age (years) | 57.8 ± 12.2 | 50 ± 11 | 59 ± 10 | 65 ± 10 |
| Male sex | 114 (53.8) | 32 (28.1) | 32 (28.1) | 50 (43.9) |
| Body Mass Index (kg/m$^2$) | 28.0 ± 4.9 | 28 ± 5 | 29 ± 5 | 27 ± 5 |
| Diabetes mellitus | 32 (15.1) | 9 (28.1) | 10 (31.3) | 13 (40.6) |
| Hypertension | 128 (60.4) | 34 (26.6) | 47 (36.7) | 47 (36.7) |
| Hypercholesterolemia | 85 (40.1) | 16 (18.8) | 34 (40.0) | 35 (41.2) |
| Smoking[a]  Current | 39 (18.4) | 17 (43.6) | 8 (20.5) | 14 (35.9) |
| Ever | 86 (40.6) | 21 (24.4) | 34 (39.5) | 31 (36.0) |
| Never | 86 (40.6) | 32 (37.2) | 28 (32.6) | 26 (30.2) |

Values represent number of subjects (%) or mean ± standard deviation.

a: For 1 patient the smoking status was unknown.

We observed that 70 (33%) patients had a zero CAC score while the remainder (n = 142) had CAC scores ranging from 1, up to 20000. This prompted us to stratify all individuals into tertiles (Table 1): group 1 containing the zero CAC scores, group 2 (n = 70, 33%) had CAC scores in the range of 1–125, and group 3 (n = 72, 34%) had CAC scores ranging from ≥126 to 6141, but also included one outlier subject with a notable CAC score of 20000.

The dental findings and the dental pathology in this study population are arranged per CAC- tertile group in Table 2. The study population consisted of 43 (20.3%) edentulous patients and 169 (79.6%) dentate patients. First we observed a significant higher percentage of edentulous patients in the higher CAC tertile (p = 0.009); there were 22 patients edentulous in the latter group, while only 9 and 12 patients in CAC tertile 1 and tertile 2 respectively. The edentulous patients in the highest CAC tertile had significant less implants (for implant retained dentures) than the patients in the lower CAC tertiles (p = 0.006); 46% of the edentulous patients in the highest CAC tertile had implants versus respectively 89% and 92% of the edentulous patients in CAC tertile 1 and 2.

169 (79.1%) patients in the study population were dentate. 152 (89.9%) of these dentate patients had only natural teeth and 17 (10.1%) of these patients had a combination of natural teeth and dental implants. The number of missing teeth per CAC tertile was significant (p = 0.03); the mean number of missing teeth was 7.6 (SD 6.6) in the lowest CAC tertile and 11.0 (SD 7.6) in the highest CAC tertile. Additionally, the number of teeth with untreated caries was significantly higher in the tertile with the highest CAC scores (p = 0.05). Furthermore, the results showed a trend for more teeth with peri-apical lesions and a higher percentage of mean alveolar bone loss in the CAC tertile group with the highest CAC scores, with a p-value of respectively 0.07 and 0.06 All other dental findings were not correlated to the CAC scores and are listed in Table 2.

**Table 2. Dental conditions.**

| | All subjects n = 212 | CAC Tertile 1 n = 70 | CAC Tertile 2 n = 70 | CAC Tertile 3 n = 72 | p-value |
|---|---|---|---|---|---|
| **Edentulous** | **43 (20.3)** | **9 (12.9)** | **12 (17.1)** | **22 (30.6)** | **0.009*◆** |
| With implants | 29 (67.4) | 8 (88.9) | 11 (91.7) | 10 (45.5) | 0.006 ◆ |
| With implants with bone loss | 7 (16.3) | 3 (33.3) | 2 (16.7) | 2 (9.1) | 0.109 |
| **Dentate** | **169 (79.6)** | **61 (87.1)** | **58 (82.9)** | **50 (69.4)** | **0.009*◆** |
| Missing teeth | 9.4 ± 7.1 | 7.6 ± 6.6 | 10.0 ± 6.8 | 11 ± 7.6 | 0.033◆ |
| Dental implants | 0.3 ± 1.0 | 0.2 ± 0.6 | 0.4 ± 1.5 | 0.3 ± 0.8 | 0.397 |
| Implants with bone loss | 0.1 ± 0.6 | 0.1 ± 0.3 | 0.2 ± 0.9 | 0.12 ± 0.4 | 0.440 |
| Restored teeth | 12.5 ± 5.5 | 11.9 ± 5.0 | 13.1 ± 5.5 | 12.7 ± 6.2 | 0.535 |
| Endodontically treated teeth | 1.9 ± 2.2 | 1.7 ± 2.0 | 1.8 ± 2.4 | 2.1 ± 2.2 | 0.640 |
| Teeth with peri-apical lesions | 2.9 ± 2.4 | 2.9 ± 2.6 | 2.4 ± 2.1 | 3.5 ± 2.4 | 0.070 |
| Teeth with caries | 3.2 ± 2.7 | 3.1 ± 3.3 | 2.7 ± 1.9 | 4.0 ± 2.7 | 0.050 |
| Radices relictae | 0.5 ± 1.2 | 0.7 ± 1.5 | 0.2 ± 0.6 | 0.6 ± 1.2 | 0.136 |
| Impacted teeth | 0.3 ± 0.7 | 0.4 ± 0.9 | 0.3 ± 0.6 | 0.3 ± 0.6 | 0.363 |
| Cysts | 0.1 ± 0.4 | 0.1 ± 0.4 | 0.2 ± 0.4 | 0.1 ± 0.3 | 0.725 |
| Mean alveolar bone loss (%) | 21.5 ± 10.7 | 20.2 ± 11.2 | 20.1 ± 9.5 | 24.4 ± 11.1 | 0.064 |

Values represent number of subjects (%) or mean ± standard deviation.

Group differences were tested with one-way ANOVA.

Tertile 1: CAC score = 0, Tertile 2: CAC score 1–125, Tertile 3: CAC score >125.

* From the same Chi-square analysis.

◆ Statistical significant, P-value <0.05.

**Table 3. Final backward linear regression model to explore variations in CAC values among 212 subjects.**

|  | B | p-value |
|---|---|---|
| Age | 0.49 | 0.000 |
| Male sex | 0.23 | 0.000 |
| Hypercholesterolemia | 0.17 | 0.004 |
| Missing teeth | 0.11 | 0.079 |

B = Standardized Beta coefficient.

Potential confounders initially included in model; Age, sex, age, BMI, diabetes, hypertension, hypercholesterolemia, smoking. Higher age, male sex and hypercholesterolemia accounted for most of the variance in CAC values.

Table 3 displays the results of modeling the CAC variation in the study population by a backwards-linear regression. Potential confounders initially included were age, sex, BMI, diabetes, hypertension, hypercholesterolemia and smoking. Tooth loss was the only dental pathology used in this model. Age, BMI and the missing teeth were continuous parameters and all other were categorical parameters. Higher age, male sex and hypercholesterolemia accounted for most of the variance in CAC values. Tooth loss had a standardized Beta correlation coefficient with the CAC scores of 0.11 (versus 0.49 for age, 0.23 for male sex and 0.17 for hypercholesterolemia) and showed a trend to be associated but this failed to reach statistical significance (p = 0.079).

## Discussion

This retrospective, cross-sectional pilot study is the first that explored the association and possible correlation between CAC scores and the common dental pathologies. The most obvious and definitive dental pathological event is tooth loss. We observed a statistically significant association between the number of teeth lost and the CAC score. However, when adjusted for age, sex and hypocholesteremia, this correlation was no longer significant (p = 0.079). Furthermore, we found univariate trends in dentate patients for an association between higher CAC scores and teeth with peri-apical lesions and untreated caries.

Tooth loss is the ultimate state of dental pathology. Most tooth loss before middle age is caused by dental caries. Dental caries is a disease with a multifactorial etiology; consumption of dietary carbohydrates is one of the most important etiological factors. Carbohydrate intake is also associated with increased risk for cardiovascular diseases and they can therefore indirectly effect each other [29]. Furthermore, tooth loss is the "end point" of periodontal disease. This prolonged state of chronic inflammation with increased levels of C-Reactive Protein (CRP) is a proven risk factor for cardiovascular diseases [14]. Besides, smokers are much more likely to develop periodontitis than non-smokers and smoking has a strong negative effect in response to periodontal treatment [11]. Smoking has therefore a well-known common effect on cardiovascular diseases and tooth loss. Above all, tooth loss might provide harmful health benefits and has been considered to impact quality of life [6, 30].

In the current study we defined the number of teeth by counting the teeth on dental panoramic radiographs (Orthopantomographs–OPGs). The number of present teeth, and correspondingly the tooth loss, is an easily accessible marker and can be determined by anyone; the general practitioner, the dentist or even the patient itself. We assumed that loss of teeth was a result of dental pathology with dental caries and periodontal disease as leading causes. This should be carefully interpreted since in some cases perhaps a tooth may have been lost due to non-pathological causes such as orthodontic treatment, dental trauma and agenesis. However, the incidence of those events is low. The OPGs were also used to determine the number of

dental implants, alveolar bone loss, caries, endodontic treatments, peri-apical radiolucencies, bone loss at implants (as a sign for peri-implantitis), impacted teeth and dental cysts. Regarding the alveolar bone loss, intra-oral radiographs are considered the standard for dental radiographic diagnostics. Nevertheless, studies have shown that OPGs and intra-oral radiographs are in great agreement [31]. For the illustration of the actual peri-apical health, a peri-apical radiograph shows a better diagnostic accuracy than an OPG [32]. Similarly, small peri-apical lesions may be better visible on intra-oral radiographs. While the OPG has a high specificity, the sensitivity is low for the detection of apical periodontitis in treated and untreated teeth, especially in the incisor area [33].

Since only radiographical and no clinical information was obtained to determine the oral health status, no assumptions could be made on the activity of the dental pathology. The observed pathology can be in an active, in a chronic or in a remission state. For example, alveolar bone loss does not necessarily accompany an active periodontitis process. Also, the peri-apical lesions could only be scored on the presence and not on activity. Peri-apical lesions can be active, inactive or a result of a healing process, i.e. a scar from previous flare-ups. However, an inactive process or a "scar" might still have caused an inflammatory process in another part of the body [34]. Previous studies in which they found a relation between (peri-apical) periodontal disease and cardiovascular diseases used clinical information [9, 13].

The maximum time allowed between the OPG and CT scan was one year. However, the average time between these two radiographic assessments was 170 days. We are aware that there is the possibility that all "scored" parameters, both for CAC scores and dental pathology, could have changed in the course of the time difference between them. We assume the pathological processes, calcium deposition as well as progression of dental pathology, are both rather slow processes and changes within 1 year will not be large. For this pilot study we deemed the maximum of 1 year acceptable.

The CAC score is used as a strong and proven biomarker for atherosclerotic cardiovascular diseases. The presence and extent of CAC can predict the presence of coronary artery stenosis, but in general it is a better marker of the extent of coronary atherosclerosis than the severity of the stenosis. However, the absence of CAC (CAC = 0) has been shown to be the strongest "negative risk factor" as compared to normal or negative values of multiple other novel risk markers for future CVD events, including carotid intima-medial thickness (CIMT), absence of carotid plaque, family history, ankle brachial index, B-type natriuretic peptide (BNP), albuminuria, family history, and hsCRP. This "power of zero" provides the strongest degree of individual "de-risking" available as compared to traditional and other novel biomarkers [35]. CAC scoring is especially useful in asymptomatic patients, but CAC also has prognostic value in symptomatic patients. However, in symptomatic patients, a CAC score of 0 does not carry the same high negative predictive value as it does in asymptomatic patients [36, 37]. In this study, the vast majority of the included patients were symptomatic.

## Conclusion

This study provides suggestive evidence that Coronary Artery Calcium is associated with the ultimate "hard" endpoint of dental pathology, i.e. tooth loss. It should be considered as a pilot study and further studies need to confirm the current findings. Nevertheless, the current findings add to the wealth of research showing the relationship between oral pathology and atherosclerotic cardiovascular diseases, in which tooth loss can be considered as an easy accessible possible marker for cardiovascular and overall health status. Health workers, especially general practitioners, dentists and cardiologists must be aware that tooth loss is sign of poor oral health and that patients with extensive tooth loss may have an increased risk for cardiovascular disease.

## Supporting information

**S1 Data.**
(SAV)

## Acknowledgments

The authors thank R.M. de Bie F. Ong, J.H. Ham, B.O. van Hamond and G. Wempe for their research work during their thesis, dr. H. Hirsch, M.O de Lange and dr. R.J. Walhout for providing their data and R.M. Brohet for his help with the statistics.

## Author Contributions

**Conceptualization:** H. C. M. Donders, L. M. IJzerman, M. Soffner, A. W. J. van 't Hof, J. de Lange.

**Data curation:** H. C. M. Donders, L. M. IJzerman, M. Soffner.

**Formal analysis:** H. C. M. Donders, L. M. IJzerman, B. G. Loos.

**Investigation:** H. C. M. Donders, L. M. IJzerman, B. G. Loos.

**Methodology:** H. C. M. Donders.

**Project administration:** H. C. M. Donders, L. M. IJzerman.

**Supervision:** B. G. Loos, J. de Lange.

**Writing – original draft:** H. C. M. Donders, L. M. IJzerman.

**Writing – review & editing:** M. Soffner, A. W. J. van 't Hof, B. G. Loos, J. de Lange.

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
