## [Decision Letter · Decision Letter 0]

21 Sep 2020

PONE-D-20-22939

Elevated Coronary Artery Calcium scores are associated with tooth loss.

PLOS ONE

Dear Dr. Donders,

Thank you for submitting your manuscript to PLOS ONE. After careful consideration, we feel that it has merit but does not fully meet PLOS ONE’s publication criteria as it currently stands. Therefore, we invite you to submit a revised version of the manuscript that addresses the points raised during the review process.

Donders et al. describe the association between dental diseases and coronary artery calcium scores. While this is interesting and original research, reviewers propose critical points to be addressed proposed. Below you will find these reviewers' comments.

We look forward to receiving your revised manuscript.

Kind regards,

Luis Cordova

Academic Editor

PLOS ONE

Journal Requirements:

2.Please provide additional details regarding participant consent. In the ethics statement in the Methods and online submission information, please ensure that you have specified (1) whether consent was informed and (2) what type you obtained (for instance, written or verbal, and if verbal, how it was documented and witnessed). If your study included minors, state whether you obtained consent from parents or guardians. If the need for consent was waived by the ethics committee, please include this information.

3.Thank you for stating the following in the Conflict of Interest and Sources of Funding Statement Section of your manuscript:

[This study is self-funded by the institutions of the authors.]

 [The author(s) received no specific funding for this work.]

Please clarify the sources of funding (financial or material support) for your study. List the grants or organizations that supported your study, including funding received from your institution.State what role the funders took in the study. If the funders had no role in your study, please state: “The funders had no role in study design, data collection and analysis, decision to publish, or preparation of the manuscript.”If any authors received a salary from any of your funders, please state which authors and which funders.

Reviewers' comments:

Reviewer's Responses to Questions

**Comments to the Author**

1. Is the manuscript technically sound, and do the data support the conclusions?

Reviewer #1: Yes

Reviewer #2: Partly

2. Has the statistical analysis been performed appropriately and rigorously? 

Reviewer #1: Yes

Reviewer #2: No

3. Have the authors made all data underlying the findings in their manuscript fully available?

Reviewer #1: Yes

Reviewer #2: No

4. Is the manuscript presented in an intelligible fashion and written in standard English?

Reviewer #1: Yes

Reviewer #2: Yes

5. Review Comments to the Author

Reviewer #1: This is an interesting and valuable manuscript that examines the correlation between age related dental pathology and coronary calcium score, which is known to be a marker for coronary artery disease and an independent predictor for myocardial infarction and sudden death. It is shown that the number of missing teeth in an older person is correlated with the coronary calcium score, and shows a stronger correlation than other markers of inflammation such as CRP. A sophisticated statistical approach (backwards analysis of variance) is used to determine whether absence of teeth is an independent predictor of a high coronary calcium score, and is not a coincidental finding attributable to coexisting conditions such as hypercholesterolemia, hypertension, smoking or male sex. The independent predictive value of tooth loss was most dramatic in patients who were edentulous. In patients with multiple missing teeth (3 to 19 missing out of the normal 32) the correlation with CAC was significant, but did not quite reach significance (P=0.076)

when adjusted for the co-variables of age, male sex and hypercholesterolemia. Several other abnormalities -- periaptical lesions and mean alveolar bone loss, were also correlated with CAC.

Examination of the mouth should be a part of every physical exam, and when the patient is edentulous, this fact is generally noted in the patient's chart. However, it is not routine for physicians internists, general practitioners or cardiologists to immediately think of coronary disease when they see a patient with who is edentulous. Moreover,

the absence of multiple missing teeth is not normally noted in the chart, or considered in the differential diagnosis

For these reasons, this article would be a useful addition to the literature. The paper is well written.

Specific comments. Smoking is certainly a significant risk factor for coronary arter disease, and is known to produce atherosclerosis, which is a cause of coronary artery calcification. Do you know if smoking has a direct harmful effect on the gums which could lead to tooth loss?

In Table II you could use symbols to highlight the P values that reach statistical signficance. For example the number of missing teeth increases with CAC in Table II and that has a significance lefel of 0.033 which is less than 0.05.

One sentence in the first paragraph of page 13 is confusing. You say "versus 89% en 92%" (sic).

Based on the comparison, it should be one value or the other. Delete the extraneous value. Also

delete "en" which is not a word and must be a typing error.

In the legend of Figure 3 you should say that "age, male sex and hypercholesterolemia accounted for most of the variance in CAC."

Reviewer #2: Donders and co-workers investigated associations between dental pathologies obtained from panoramic radiographs and coronary artery calcium scores obtained by CT scans. The study is interesting and describes novel aspects. However, details of analyses are inadequately described, which makes the judgement of the quality difficult.

Detailed comments:

-Abstract, write out ACVD

-CVD risk factors here include hypercholesterolemia, hypertension and diabetes. Were the diagnoses based on treatment and were all patients treated?

-Table 2. Give the statistical test in the footnotes. Write out PA lesions.

-Consider adding a histogram of the CAC parameter.

-Table 3. It is not clear, how the final regression model was calculated. Did you include all dental parameters in the same model? Which parameters were categorical / continuous? Many of them are strongly correlated with each other. What is meant by ‘the mean number of dental pathologies’?

-Did you do power calculations before the work, is the statistical power of this limited population adequate?

-Considering the readers of the journal, you might consider revising the Discussion, it is a bit uninteresting and contains a lot of details on radiographic techniques and their comparisons.

-Missing teeth as a predicting parameter of incident CVD has been published and it could be cited (Liljestrand JM, 2015).

6. PLOS authors have the option to publish the peer review history of their article (what does this mean?). If published, this will include your full peer review and any attached files.

Reviewer #1: No

Reviewer #2: No

---

## [Author Response · Author response to Decision Letter 0]

3 Nov 2020

Dear Luis Cordova,

We were pleased to have the opportunity to revise our manuscript entitled “Elevated coronary artery calcium scores are associated with tooth loss”. The reviewer’s comments were very helpful overall and we are appreciative of such constructive feedback to our original submission. In the revised manuscript we have carefully considered the editor and reviewers suggestions and we adjusted and added text accordingly. In the rest of the letter we address and respond to each point raised by the reviewers. The responses to the reviewers are below and are color-coded as follows: Comment from editors or reviewers are colored in red and our response is shown under each comment as normal text with a referral to the manuscript in blue. 

On behalf of my co-authors, I would thank you and the revisers for your effort and time.

Best regards 

Marie-Chris Donders

Response to editor’s comments: 

 We’ve checked the style requirements again and the manuscript is now according to the PLOS ONE style templates

2.Please provide additional details regarding participant consent. In the ethics statement in the Methods and online submission information, please ensure that you have specified (1) whether consent was informed and (2) what type you obtained (for instance, written or verbal, and if verbal, how it was documented and witnessed). If your study included minors, state whether you obtained consent from parents or guardians. If the need for consent was waived by the ethics committee, please include this information.

 The Medical Ethical Committee waived the requirement for informed consent.

We’ve added this information to the Material and Methods section on page 7 as follows:

Materials and Methods

For this retrospective cross-sectional study, data were collected from three hospitals on different locations in the Netherlands (Academic Medical Centre, Amsterdam; Isala Hospital, Zwolle; Ziekenhuis Gelderse Vallei, Ede). Patients were included when the hospital data provided a CAC score and a dental panoramic radiograph (Orthopantomograph – OPG), both obtained within a maximum period of 365 days between them, from 2009-2017. All data were anonymized before assessed. This study was approved by the Medical Ethical Committee (15.06107) of the Isala Hospital, Zwolle and accepted by the other participating hospitals. The Medical Ethical Committee waived the requirement for informed consent.

3.Thank you for stating the following in the Conflict of Interest and Sources of Funding Statement Section of your manuscript:

[This study is self-funded by the institutions of the authors.]

 [The author(s) received no specific funding for this work.]

a. Please clarify the sources of funding (financial or material support) for your study. List the grants or organizations that supported your study, including funding received from your institution.

We’ve removed the “Conflict of Interest and Sources of Funding Statement” of the title page of the manuscript and we ask you kindly to change the online submission form for us as follows:

“ The authors received no specific funding for this work”.

 

Response to reviewer #1:

Reviewer #1: This is an interesting and valuable manuscript that examines the correlation between age related dental pathology and coronary calcium score, which is known to be a marker for coronary artery disease and an independent predictor for myocardial infarction and sudden death. It is shown that the number of missing teeth in an older person is correlated with the coronary calcium score, and shows a stronger correlation than other markers of inflammation such as CRP. A sophisticated statistical approach (backwards analysis of variance) is used to determine whether absence of teeth is an independent predictor of a high coronary calcium score, and is not a coincidental finding attributable to coexisting conditions such as hypercholesterolemia, hypertension, smoking or male sex. The independent predictive value of tooth loss was most dramatic in patients who were edentulous. In patients with multiple missing teeth (3 to 19 missing out of the normal 32) the correlation with CAC was significant, but did not quite reach significance (P=0.076)

when adjusted for the co-variables of age, male sex and hypercholesterolemia. Several other abnormalities -- periaptical lesions and mean alveolar bone loss, were also correlated with CAC.

Examination of the mouth should be a part of every physical exam, and when the patient is edentulous, this fact is generally noted in the patient's chart. However, it is not routine for physicians internists, general practitioners or cardiologists to immediately think of coronary disease when they see a patient with who is edentulous. Moreover,

the absence of multiple missing teeth is not normally noted in the chart, or considered in the differential diagnosis

For these reasons, this article would be a useful addition to the literature. The paper is well written.

Specific comments. Smoking is certainly a significant risk factor for coronary arter disease, and is known to produce atherosclerosis, which is a cause of coronary artery calcification. Do you know if smoking has a direct harmful effect on the gums which could lead to tooth loss?

Thank you for this specific comment. Smoking has indeed a negative effect in the gums, which can lead to tooth loss. We have amended the manuscript as follows in the discussion part on page 16.

Tooth loss is the ultimate state of dental pathology. Most tooth loss before middle age is caused by dental caries. Dental caries is a disease with a multifactorial etiology; consumption of dietary carbohydrates is one of the most important etiological factors. Carbohydrate intake is also associated with increased risk for cardiovascular diseases and they can therefore indirectly effect each other [28]. Furthermore, tooth loss is the “end point” of periodontal disease. This prolonged state of chronic inflammation with increased levels of C-Reactive Protein (CRP) is a proven risk factor for cardiovascular diseases [13]. Besides, smokers are much more likely to develop periodontitis than non-smokers and smoking has a strong negative effect in response to periodontal treatment [10]. Smoking has therefore a well-known common effect on cardiovascular diseases and tooth loss. Above all, tooth loss might provide harmful health benefits and has been considered to impact quality of life [5][29].

In Table II you could use symbols to highlight the P values that reach statistical significance. For example the number of missing teeth increases with CAC in Table II and that has a significance level of 0.033 which is less than 0.05. Table II is amended follows on page 14.

Table 2. Dental conditions

 All subjects

n=212 CAC

Tertile 1

n=70 CAC

Tertile 2

n=70 CAC

Tertile 3

n=72 p-value

Edentulous 43 (20.3) 9 (12.9) 12 (17.1) 22 (30.6) 0.009*

With implants 29 (67.4) 8 (88.9) 11 (91.7) 10 (45.5) 0.006 �

With implants with bone loss 7 (16.3) 3 (33.3) 2 (16.7) 2 (9.1) 0.109

Dentate 169 (79.6) 61 (87.1) 58 (82.9) 50 (69.4) 0.009*

Missing teeth 9.4 ± 7.1 7.6 ± 6.6 10.0 ± 6.8 11 ± 7.6 0.033

Dental implants 0.3 ± 1.0 0.2 ± 0.6 0.4 ± 1.5 0.3 ± 0.8 0.397

Implants with bone loss 0.1 ± 0.6 0.1 ± 0.3 0.2 ± 0.9 0.12 ± 0.4 0.440

Restored teeth 12.5 ± 5.5 11.9 ± 5.0 13.1 ± 5.5 12.7 ± 6.2 0.535

Endodontically treated teeth 1.9 ± 2.2 1.7 ± 2.0 1.8 ± 2.4 2.1 ± 2.2 0.640

Teeth with peri-apical lesions 2.9 ± 2.4 2.9 ± 2.6 2.4 ± 2.1 3.5 ± 2.4 0.070

Teeth with caries 3.2 ± 2.7 3.1 ± 3.3 2.7 ± 1.9 4.0 ± 2.7 0.050

Radices relictae 0.5 ± 1.2 0.7 ± 1.5 0.2 ± 0.6 0.6 ± 1.2 0.136

Impacted teeth 0.3 ± 0.7 0.4 ± 0.9 0.3 ± 0.6 0.3 ± 0.6 0.363

Cysts 0.1 ± 0.4 0.1 ± 0.4 0.2 ± 0.4 0.1 ± 0.3 0.725

Mean alveolar bone loss (%) 21.5 ± 10.7 20.2 ± 11.2 20.1 ± 9.5 24.4 ± 11.1 0.064

Values represent number of subjects (%) or mean ± standard deviation

Group differences were tested with one-way ANOVA

Tertile 1: CAC score = 0, Tertile 2: CAC score 1-125, Tertile 3: CAC score >125

* From the same Chi-square analysis

Statistical significant, P-value <0.05

One sentence in the first paragraph of page 13 is confusing. You say "versus 89% en 92%" (sic). Based on the comparison, it should be one value or the other. Delete the extraneous value. Also delete "en" which is not a word and must be a typing error.

We’ve amended this sentence on page 13 as follows:

The edentulous patients in the highest CAC tertile had significant less implants (for implant retained dentures) than the patients in the lower CAC tertiles (p=0.006); 46% of the edentulous patients in the highest CAC tertile had implants versus respectively 89% and 92% of the edentulous patients in CAC tertile 1 and 2.

In the legend of Figure 3 you should say that "age, male sex and hypercholesterolemia accounted for most of the variance in CAC."

We’ve added this comment to the legend and the accompanying text to table 3 on page 15 as follows: 

Table 3 displays the results of modeling the CAC variation in the study population by a backwards-linear regression. Potential confounders initially included were age, sex, BMI, diabetes, hypertension, hypercholesterolemia and smoking. Tooth loss was the only dental pathology used in this model. Age, BMI and the missing teeth were continuous parameters and all other were categorical parameters. Higher age, male sex and hypercholesterolemia accounted for most of the variance in CAC values. Tooth loss had a standardized Beta correlation coefficient with the CAC scores of 0.11 (versus 0.49 for age, 0.23 for male sex and 0.17 for hypercholesterolemia) and showed a trend to be associated but this failed to reach statistical significance (p=0.079).

 

Response to reviewer #2:

Reviewer #2: Donders and co-workers investigated associations between dental pathologies obtained from panoramic radiographs and coronary artery calcium scores obtained by CT scans. The study is interesting and describes novel aspects. However, details of analyses are inadequately described, which makes the judgement of the quality difficult.

Detailed comments:

-Abstract, write out ACVD

We’ve amended this sentence in the abstract on page 3 as follows:

The number of (missing) teeth is an easy accessible marker and could be used as a marker for atherosclerotic cardiovascular disease (ACVD) risk by almost any healthcare worker.

-CVD risk factors here include hypercholesterolemia, hypertension and diabetes. Were the diagnoses based on treatment and were all patients treated?

We’ve amended the “patient characteristics” section on page 7 as follows:

Patient characteristics

The electronic health records provided information about: sex, age, smoking, diabetes mellitus, hypercholesterolemia, hypertension and body mass index (BMI). When diabetes mellitus, hypercholesterolemia or hypertension were not mentioned in a patient file, but the corresponding medication was available (e.g. metformin and/or insulin, statins and antihypertensive drugs), the patient was scored positively for that disorder. BMI was calculated with the noted height and weight on the day of the CT-scan for the CAC score.

-Table 2. Give the statistical test in the footnotes. Write out PA lesions.

Table 2 is amended conform comments on page 14 as follows:

Values represent number of subjects (%) or mean ± standard deviation

Group differences were tested with one-way ANOVA

Tertile 1: CAC score = 0, Tertile 2: CAC score 1-125, Tertile 3: CAC score >125

* From the same Chi-square analysis

Statistical significant, P-value <0.05

-Consider adding a histogram of the CAC parameter.

First, we considered a boxplot of the CAC scores, but finally decided not to use it, because in our opinion did not add added extra value to the data in the table.

-Table 3. It is not clear, how the final regression model was calculated. Did you include all dental parameters in the same model? Which parameters were categorical / continuous? Many of them are strongly correlated with each other. 

We’ve added this comment to the accompanying text to table 3 on page 15 as follows: 

Table 3 displays the results of modeling the CAC variation in the study population by a backwards-linear regression. Potential confounders initially included were age, sex, BMI, diabetes, hypertension, hypercholesterolemia and smoking. Tooth loss was the only dental pathology used in this model. Age, BMI and the missing teeth were continuous parameters and all other were categorical parameters. Higher age, male sex and hypercholesterolemia accounted for most of the variance in CAC values. Tooth loss had a standardized Beta correlation coefficient with the CAC scores of 0.11 (versus 0.49 for age, 0.23 for male sex and 0.17 for hypercholesterolemia) and showed a trend to be associated but this failed to reach statistical significance (p=0.079).

What is meant by ‘the mean number of dental pathologies’?

We’ve amended the sentence on page 10 as follows, to clarify:

The mean numbers of the dental pathologies scored on the OPG were calculated per group and possible differences between groups were tested by ANOVA

-Did you do power calculations before the work, is the statistical power of this limited population adequate?

There were no previous data available for a proper power calculation., therefore the current study is considered a pilot study. 

-Considering the readers of the journal, you might consider revising the Discussion, it is a bit uninteresting and contains a lot of details on radiographic techniques and their comparisons.

We’ve added extra interesting text and some radiographic details were removed in the discussion part. We want to reach dentists, cardiologists and especially more general doctors and health workers with this paper.

Discussion

This retrospective, cross-sectional pilot study is the first that explored the association and possible correlation between CAC scores and the common dental pathologies. The most obvious and definitive dental pathological event is tooth loss. We observed a statistically significant association between the number of teeth lost and the CAC score. However, when adjusted for age, sex and hypocholesteremia, this correlation was no longer significant (p=0.079). Furthermore, we found univariate trends in dentate patients for an association between higher CAC scores and teeth with peri-apical lesions and untreated caries. 

Tooth loss is the ultimate state of dental pathology. Most tooth loss before middle age is caused by dental caries. Dental caries is a disease with a multifactorial etiology; consumption of dietary carbohydrates is one of the most important etiological factors. Carbohydrate intake is also associated with increased risk for cardiovascular diseases and they can therefore indirectly effect each other [28]. Furthermore, tooth loss is the “end point” of periodontal disease. This prolonged state of chronic inflammation with increased levels of C-Reactive Protein (CRP) is a proven risk factor for cardiovascular diseases [13]. Besides, smokers are much more likely to develop periodontitis than non-smokers and smoking has a strong negative effect in response to periodontal treatment [10]. Smoking has therefore a well-known common effect on cardiovascular diseases and tooth loss. Above all, tooth loss might provide harmful health benefits and has been considered to impact quality of life [5][29].

In the current study we defined the number of teeth by counting the teeth on dental panoramic radiographs (Orthopantomographs – OPGs). The number of present teeth, and correspondingly the tooth loss, is an easily accessible marker and can be determined by anyone; the general practitioner, the dentist or even the patient itself. We assumed that loss of teeth was a result of dental pathology with dental caries and periodontal disease as leading causes. This should be carefully interpreted since in some cases perhaps a tooth may have been lost due to non-pathological causes such as orthodontic treatment, dental trauma and agenesis. However, the incidence of those events is low. The OPGs were also used to determine the number of dental implants, alveolar bone loss, caries, endodontic treatments, peri-apical radiolucencies, bone loss at implants (as a sign for peri-implantitis), impacted teeth and dental cysts. Regarding the alveolar bone loss, intra-oral radiographs are considered the standard for dental radiographic diagnostics. Nevertheless, studies have shown that OPGs and intra-oral radiographs are in great agreement [30]. For the illustration of the actual peri-apical health, a peri-apical radiograph shows a better diagnostic accuracy than an OPG [31]. Similarly, small peri-apical lesions may be better visible on intra-oral radiographs. While the OPG has a high specificity, the sensitivity is low for the detection of apical periodontitis in treated and untreated teeth, especially in the incisor area [32][33]. 

Since only radiographical and no clinical information was obtained to determine the oral health status, no assumptions could be made on the activity of the dental pathology. The observed pathology can be in an active, in a chronic or in a remission state. For example, alveolar bone loss does not necessarily accompany an active periodontitis process. Also, the peri-apical lesions could only be scored on the presence and not on activity. Peri-apical lesions can be active, inactive or a result of a healing process, i.e. a scar from previous flare-ups. However, an inactive process or a “scar” might still have caused an inflammatory process in another part of the body [34]. Previous studies in which they found a relation between (peri-apical) periodontal disease and cardiovascular diseases used clinical information [9][13].

The maximum time allowed between the OPG and CT scan was one year. However, the average time between these two radiographic assessments was 170 days. We are aware that there is the possibility that all “scored” parameters, both for CAC scores and dental pathology, could have changed in the course of the time difference between them. We assume the pathological processes, calcium deposition as well as progression of dental pathology, are both rather slow processes and changes within 1 year will not be large. For this pilot study we deemed the maximum of 1 year acceptable.

The CAC score is used as a strong and proven biomarker for atherosclerotic cardiovascular diseases. The presence and extent of CAC can predict the presence of coronary artery stenosis, but in general it is a better marker of the extent of coronary atherosclerosis than the severity of the stenosis. However, the absence of CAC (CAC = 0) has been shown to be the strongest "negative risk factor" as compared to normal or negative values of multiple other novel risk markers for future CVD events, including carotid intima-medial thickness (CIMT), absence of carotid plaque, family history, ankle brachial index, B-type natriuretic peptide (BNP), albuminuria, family history, and hsCRP. This "power of zero" provides the strongest degree of individual "de-risking" available as compared to traditional and other novel biomarkers [35]. CAC scoring is especially useful in asymptomatic patients, but CAC also has prognostic value in symptomatic patients. However, in symptomatic patients, a CAC score of 0 does not carry the same high negative predictive value as it does in asymptomatic patients. [36][37]. In this study, the vast majority of the included patients were symptomatic.

-Missing teeth as a predicting parameter of incident CVD has been published and it could be cited (Liljestrand JM, 2015).

A citation to this interesting article of Liljestrand JM et.al is added to the introduction part on page 4 as follows:

Tooth loss is the ultimate event representing dental pathologies. Various articles have found that missing teeth are predictors of incident CVD, for example as was reported by Liljestrand et al.[5] More recently an interesting meta-analysis on tooth loss and risk of ACVD and stroke was published [6].

---

## [Decision Letter · Decision Letter 1]

18 Nov 2020

Elevated Coronary Artery Calcium scores are associated with tooth loss.

PONE-D-20-22939R1

Dear Dr. Marie-Chris Donders,

We’re pleased to inform you that your manuscript has been judged scientifically suitable for publication and will be formally accepted for publication once it meets all outstanding technical requirements.

Kind regards,

Luis Cordova

Academic Editor

PLOS ONE

Additional Editor Comments (optional):

The authors have addressed previous reviewer's comments increasing the clarity and value of this research.

Reviewers' comments:

Reviewer's Responses to Questions

**Comments to the Author**

1. If the authors have adequately addressed your comments raised in a previous round of review and you feel that this manuscript is now acceptable for publication, you may indicate that here to bypass the “Comments to the Author” section, enter your conflict of interest statement in the “Confidential to Editor” section, and submit your "Accept" recommendation.

Reviewer #2: (No Response)

2. Is the manuscript technically sound, and do the data support the conclusions?

Reviewer #2: (No Response)

3. Has the statistical analysis been performed appropriately and rigorously? 

Reviewer #2: (No Response)

4. Have the authors made all data underlying the findings in their manuscript fully available?

Reviewer #2: (No Response)

5. Is the manuscript presented in an intelligible fashion and written in standard English?

Reviewer #2: (No Response)

6. Review Comments to the Author

Reviewer #2: Thank you for the carefully done revision of the manuscriopt. The authors have adequately addressed my comments raised in a previous round of review and I do not have further criticism.

7. PLOS authors have the option to publish the peer review history of their article (what does this mean?). If published, this will include your full peer review and any attached files.

Reviewer #2: No

---

## [Editor Report · Acceptance letter]

24 Nov 2020

PONE-D-20-22939R1 

Elevated Coronary Artery Calcium scores are associated with tooth loss. 

Dear Dr. Donders:

I'm pleased to inform you that your manuscript has been deemed suitable for publication in PLOS ONE. Congratulations! Your manuscript is now with our production department. 

Kind regards, 

on behalf of

Dr. Luis Cordova 

Academic Editor

PLOS ONE